# Risk-Averse Offline Reinforcement Learning

**Núria Armengol Urpí**
Dept. of Computer Science
ETH Zurich
narmengolurpi@gmail.com

**Sebastian Curi**
Dept. of Computer Science
ETH Zurich
scuri@inf.ethz.ch

**Andreas Krause**
Dept. of Computer Science
ETH Zurich
krausea@ethz.ch

## Abstract

Training Reinforcement Learning (RL) agents online in high-stakes applications is often prohibitive due to the risk associated with exploration. Thus, the agent can only use data previously collected by *safe* policies. While previous work considers optimizing the *average* performance using offline data, we focus on optimizing a *risk-averse* criterion. In particular, we present the *Offline Risk-Averse Actor-Critic* (O-RAAC), a model-free RL algorithm that is able to learn risk-averse policies in a fully offline setting. We show that O-RAAC learns policies with higher risk-averse performance than risk-neutral approaches in different robot control tasks. Furthermore, considering risk-averse criteria guarantees distributional robustness of the average performance with respect to particular distribution shifts. We demonstrate empirically that in the presence of natural distribution-shifts, O-RAAC learns policies with good average performance.

## 1 Introduction

In high-stakes applications, the deployment of highly-performing Reinforcement Learning (RL) agents is limited by prohibitively large costs at early exploration stages (Dulac-Arnold et al., 2019). To address this issue, the offline (or batch) RL setting considers learning a policy from a limited batch of pre-collected data. However, high-stakes decision-making is typically also *risk-averse*: we assign more weight to adverse events than to positive ones (Pratt, 1978). Although several algorithms for risk-sensitive RL exist (Howard & Matheson, 1972; Mihatsch & Neuneier, 2002), none of them addresses the offline setting. On the other hand, existing offline RL algorithms consider the *average* performance criterion and are risk-neutral (Ernst et al., 2005; Lange et al., 2012).

**Main contributions**  We present the first approach towards *learning a risk-averse RL policy for high-stakes applications using only offline data*: the **O**ffline **R**isk-**A**verse **A**ctor-**C**ritic (O-RAAC). The algorithm has three components: a distributional critic that learns the full value distribution (Section 3.1), a risk-averse actor that optimizes a risk averse criteria (Section 3.2) and an imitation learner implemented with a variational auto-encoder (VAE) that reduces the bootstrapping error due to the offline nature of the algorithm (Section 3.3). In Figure 1, we show how these components interact with each other. Finally, in Section 4 we demonstrate the empirical performance of O-RAAC. Our implementation is freely available at Github: https://github.com/nuria95/O-RAAC.

### 1.1 Related Work

**Risk-Averse RL**  The most common risk-averse measure in the literature is the Conditional Value-at-Risk (CVaR) (Rockafellar & Uryasev, 2002), which corresponds to the family of Coherent Risk-Measures (Artzner et al., 1999), and we focus mainly on these risk-measures. Nevertheless, other risk criteria such as Cumulative Prospect Theory (Tversky & Kahneman, 1992) or Exponential Utility (Rabin, 2013) can also be used with the algorithm we propose. In the context of RL, Petrik & Subramanian (2012); Chow & Ghavamzadeh (2014); Chow et al. (2015) propose dynamic programming algorithms for solving the CVaR of the return distribution with *known* tabular Markov Decision Processes (MDPs). For unknown models, Morimura et al. (2010) propose a SARSA algorithm for (CVaR) optimization but it is limited to the on-policy setting and small action spaces. To scale to larger systems, Tamar et al. (2012; 2015) propose on-policy Actor-Critic algorithms for Coherent Risk-Measures. However, they are extremely sample inefficient due to sample discarding to compute

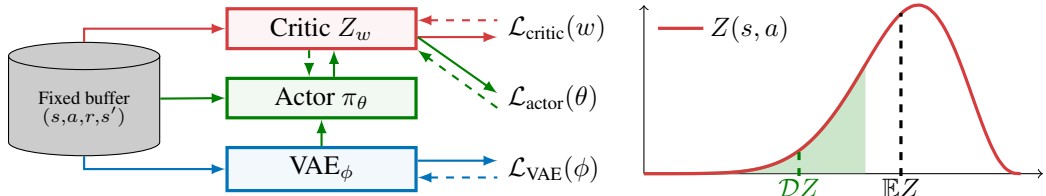

Figure 1: Visualization of the algorithm components. Solid lines indicate the forward flow of data whereas dashed lines indicate the backward flow of gradients. Data is stored in the fixed buffer. The VAE, in blue, learns a generative model of the behavior policy. The actor, in green, perturbs the VAE and outputs a policy. The critic, learns the $Z$-value distribution of the policy. The actor optimizes a risk-averse distortion of the $Z$-value distribution, which we denote by $\mathcal{D}Z$. On the right, we show a typical probability density function of $Z$ learned by the critic in red. In dashed black we indicate the expected value of $Z$, which a risk-neutral actor intends to maximize. Instead, a risk-averse actor intends to maximize a distortion $\mathcal{D}Z$, shown in dashed green. In this particular visualization, we show the ubiquitous Conditional Value at Risk (CVaR).

the risk-criteria and the high-variance of the gradient estimate. While Prashanth et al. (2016) address sample efficiency by considering Cumulative Prospect Theory instead of Coherent Risk-Measures, their algorithm is limited to tabular MDPs and is also on-policy. Instead, Tang et al. (2020) propose an off-policy algorithm that approximates the return distribution with a Gaussian distribution and learns its moments using the Bellman equation for the mean and the variance of the distribution. Instead, we learn the full return distribution without making the Gaussianity assumption (Bellemare et al., 2017). Perhaps most closely related is the work of Singh et al. (2020), who consider also a distributional critic but their algorithm is limited to the CVaR and they do not address the offline RL setting. Furthermore, they use a sample-based distributional critic, which makes the computation of the CVaR inefficient. Instead, we modify Implicit Quantile Networks (Dabney et al., 2018) in order to compute different risk criteria efficiently. Although (Dabney et al., 2018) already investigated risk-related criteria, their scope is limited to discrete action spaces (e.g., the Atari domain) in an off-policy setting whereas we consider continuous actions in an offline setting.

**Offline RL** The biggest challenge in offline RL is the Bootstrapping Error: a Q-function is evaluated at state-action pairs where there is little or no data and these get propagated through the Bellman equation (Kumar et al., 2019). In turn, a policy optimized with offline data induces a state-action distribution that is shifted from the original data (Ross et al., 2011). To address this, Fujimoto et al. (2019) propose to express the actor as the sum between an imitation learning component and a perturbation model to control the deviation of the behavior policy. Other approaches to control the difference between the data-collection policy and the optimized policy include regularizing the policies with the behavior policy using the MMD distance (Kumar et al., 2019) or f-divergences (Wu et al., 2020; Jaques et al., 2019), or using the behavior policy as a prior (Siegel et al., 2020). An alternative strategy in offline RL is to be *pessimistic* with respect to the epistemic uncertainty that arises due to data scarcity. Yu et al. (2020) take a model-based approach and penalize the per-step rewards with the epistemic uncertainty of their dynamical model. Using a model-free approach Kumar et al. (2020); Buckman et al. (2020) propose to learn a lower bound of the Q-function using an estimate of the uncertainty as penalty in the target of the equation. Our work uses ideas from both strategies to address the offline risk-averse problem. First, we use an imitation learner to control the bootstrapping error. However, by considering a risk-averse criterion, we are also optimizing over a *pessimistic* distribution compatible with the empirical distribution in the data set. The connections between risk-aversion and distributional robustness are well studied in supervised learning (Shapiro et al., 2014; Namkoong & Duchi, 2017; Curi et al., 2020; Levy et al., 2020) and in reinforcement learning (Chow et al., 2015; Pan et al., 2019).

## 2 PROBLEM STATEMENT

We consider a Markov Decision Process (MDP) with possibly continuous $s \in \mathcal{S}$ and possibly continuous actions $a \in \mathcal{A}$, transition kernel $P(\cdot|s, a)$, reward kernel $R(\cdot|s, a)$ and discount factor $\gamma$. We denote by $\pi$ a stationary policy, i.e., a mapping from states to distribution over actions. We have

access to a fixed batch data set collected with an unknown behaviour policy $\pi_\beta$. We call $d^\beta$ the joint *state, action, reward, next-state* distribution induced by the behaviour policy and $\rho^\beta$ the marginal *state* distribution. We access this distribution by sampling from the fixed data set. Similarly, for any policy $\pi$, we call $d^\pi$ the joint *state, action, reward, next-state* distribution induced by $\pi$ on the MDP.

In risk-neutral RL, the goal is to find a policy that maximizes the *expected* discounted sum of returns $\mathbb{E}_{d^\pi}\left[\sum_{t=1}^\infty \gamma^{t-1} R(\cdot|S_t, A_t)\right]$, where the expectation is taken with respect to the stochasticity introduced by the reward kernel, the transition kernel, and the policy. We define as $Z^\pi(s, a) =_D \sum_{t=1}^\infty \gamma^{t-1} R(\cdot|S_t, A_t)$ as the return distribution conditioned on $(S_1 = s, A_1 = a)$ and following $\pi$ thereafter. Here $=_D$ denotes equality in distribution. The risk-neutral RL objective is the expectation of the distribution of $Z^\pi$.

In risk-averse settings, we replace the expectation with a distortion operator $\mathcal{D}$ that is a mapping from the distribution over the returns to the reals. Thus, the goal is to find a policy $\pi$ that maximizes

$$\max_\pi \mathcal{D}\left[\sum_{t=1}^\infty \gamma^{t-1} R(\cdot|S_t, A_t)\right]. \tag{1}$$

With this framework, we address many different risk-averse distortions. For example, this includes probability weighting functions (Gonzalez & Wu, 1999; Tversky & Kahneman, 1992), the CVaR (Rockafellar & Uryasev, 2002), the mean-variance criteria (Namkoong & Duchi, 2017) or the Wang criteria (Wang, 1996; Rabin, 2013).

## 3 OFFLINE RISK-AVERSE ACTOR-CRITIC (O-RAAC)

We now present our algorithm O-RAAC for offline-risk averse RL. One of the main technical challenges in going beyond expected rewards is to find an analogue of the Q-function for the particular distortion operator we want to optimize. Unfortunately, the Bellman target of most risk-analogues does not have a closed-form expression. Therefore, we instead learn the full distribution of returns as proposed by Bellemare et al. (2017). In Section 3.1, we describe the training procedure for the distributional critic. Next, in Section 3.2 we define the actor loss as the risk-distortion operator on the learned return distribution and optimize it using a gradient-based approach. Up to this point, the actor-critic template is enough to optimize a risk-averse criteria. However, as we focus on the offline setting, we need to control the bootstrapping error. To this end, we use a variational auto-encoder VAE to learn a generative model of the behavior policy in Section 3.3. Finally, in Section 3.4 we bring all the pieces together and instantiate our algorithm for different risk distortions $\mathcal{D}$.

### 3.1 DISTRIBUTIONAL CRITIC LEARNING

To learn the distributional critic, we exploit the distributional Bellman equation of returns $Z^\pi(s, a) =_D R(s, a) + \gamma Z^\pi(S', A')$ for policy evaluation. The random variables $S', A'$ are distributed according to $s' \sim P(\cdot|s, a)$ and $A' \sim \pi(\cdot|s')$. In particular, we represent the return distribution implicitly through its quantile function as proposed by Dabney et al. (2018). We use this representation because many risk distortion operators can be efficiently computed using the quantile function of the underlying random variable. We parameterize the quantile function through a neural network with learnable parameters $w$. We express such implicit quantile function as $Z_w^\pi(s, a; \tau)$, where $\tau \in [0, 1]$ is the quantile level. Whereas the neural network architecture proposed by Dabney et al. (2018) is for discrete actions only, we extend it to continuous actions by considering all $s, a$, and $\tau$ as the inputs and only the quantile value as the output. To learn the parameters $w$, we use the distributional variant of fitted value-iteration (Bellemare et al., 2017; Munos & Szepesvári, 2008) using a quantile Huber-loss (Huber, 1964) as a surrogate of the Wasserstein-distance as proposed by Dabney et al. (2018). To this end, we use a target network with parameters $w'$ and compute the temporal difference (TD) error at a sample $(s, a, r, s')$ as

$$\delta_{\tau,\tau'} = r + \gamma Z_{w'}^\pi(s', a'; \tau') - Z_w^\pi(s, a; \tau), \tag{2}$$

for $\tau, \tau'$ independently sampled from the uniform distribution, i.e., $\tau, \tau' \sim \mathcal{U}(0, 1)$ and $a' \sim \pi(\cdot|s')$. The $\tau$-quantile Huber-loss is

$$\mathcal{L}_\kappa(\delta; \tau) = \underbrace{\left|\tau - \mathbb{1}_{\{\delta<0\}}\right|}_{\text{Quantile loss}} \cdot \underbrace{\begin{cases} \frac{1}{2\kappa}\delta^2 & \text{if } |\delta| \leq \kappa, \\ |\delta| - \frac{1}{2}\kappa & \text{otherwise.} \end{cases}}_{\text{Huber loss}} \tag{3}$$

We prefer the Huber loss over the $\mathcal{L}_2$ or $\mathcal{L}_1$ loss as it is better behaved due to smooth gradient-clipping (Mnih et al., 2015). Finally, we approximate the quantile loss for all levels $\tau$ by sampling $N$ independent quantiles $\tau$ and $N'$ independent target quantiles $\tau'$. The critic loss is

$$\mathcal{L}_{\text{critic}}(w) = \mathbb{E}_{\substack{(s,a,r,s') \sim d^\beta(\cdot) \\ a' \sim \pi(\cdot|s')}} \Big[ \frac{1}{N \cdot N'} \sum_{i=1}^{N} \sum_{j=1}^{N'} \mathcal{L}_\kappa(\delta_{\tau_i, \tau_j'}; \tau_i) \Big]. \tag{4}$$

## 3.2 LEARNING A RISK-AVERSE ACTOR

In risk-averse applications, we generally prefer deterministic policies over stochastic ones because introducing extra randomness is against a risk-averse behavior (Pratt, 1978) and there is no benefit of exploration often associated with stochastic policies in the offline setting. Hence, we consider parameterized deterministic policies $\pi_\theta(s) : \mathcal{S} \to \mathcal{A}$. Given a learned distributional critic, we define the actor loss as

$$\mathcal{L}_{\text{actor}}(\theta) = -\mathbb{E}_{s \sim \rho^\beta(\cdot)} \left[ \mathcal{D} \left( Z_w^{\pi_\theta}(s, \pi_\theta(s); \tau) \right) \right], \tag{5}$$

where $\mathcal{D}$ is the operator that models risk aversion. We consider Markovian policies because these contain the optimal policy for many common risk distortions in the discounted infinite horizon setting (Ruszczyński, 2010, Theorem 4). To minimize the actor loss (5), we use pathwise derivatives of the objective (Mohamed et al., 2020), computed by backpropagating the actor through the learned critic at states sampled from the offline data set. Minimizing the actor loss is equivalent to maximizing the risk-averse performance.

We leverage the quantile representation of the critic (c.f. Section 3.1) to compute the risk distortion operator inside the actor loss (5), Given a quantile representation of a distribution, common risk distortions $\mathcal{D}$ can be efficiently approximated using a sampling-based scheme. In particular, there exists a quantile sampling distribution $\mathbb{P}_\mathcal{D}$ associated to $\mathcal{D}$ such that

$$\mathcal{D} \left( Z_w^{\pi_\theta}(s, \pi_\theta(s); \tau) \right) = \int Z_w^{\pi_\theta}(s, \pi_\theta(s); \tau) \mathbb{P}_\mathcal{D}(\tau) \, \mathrm{d}\tau \approx \frac{1}{K} \sum_{k=1}^{K} Z_w^{\pi_\theta}(s, \pi_\theta(s); \tau_k), \ \tau_k \sim \mathbb{P}_\mathcal{D}. \tag{6}$$

For cumulative prospect theory (Tversky & Kahneman, 1992) and coherent risk measures (Artzner et al., 1999), the associated quantile sampling distributions are well-known. In the particular case of the CVaR, this is known as the Acerbi's formula (Acerbi & Tasche, 2002)

$$\text{CVaR}_\alpha(Z_w^{\pi_\theta}(s, a; \tau)) = \frac{1}{\alpha} \int_0^\alpha Z_w^{\pi_\theta}(s, a; \tau) \, \mathrm{d}\tau. \tag{7}$$

When other representations are used for the critic, computing the risk-distortion $\mathcal{D}$ becomes computationally expensive. In some particular cases, there are variational formulas to compute the risk-distortion $\mathcal{D}$ that require to solve an inner optimization problem. For example, Singh et al. (2020) use the common Rockafellar truncated optimization procedure (Rockafellar & Uryasev, 2002) to compute the CVaR, which is sample inefficient and has high variance (Curi et al., 2020).

## 3.3 OFF-POLICY TO OFFLINE: CONTROLLING THE BOOTSTRAPING ERROR.

Up to this point, the actor-critic procedure we describe in Sections 3.1 and 3.2 is theoretically sufficient to learn a risk-averse RL agent. However, in the offline setting the bootstrapping error appears: when evaluating the TD-error (2), the $Z$-value target will be evaluated at actions where there is no data (Kumar et al., 2019) and propagated through the Bellman equation. To address this issue, (Kumar et al., 2019; Wu et al., 2020; Siegel et al., 2020) propose stochastic policies and penalize deviations from the behaviour policy while optimizing the actor.

As discussed in Section 3.2, on one hand we prefer deterministic policies over stochastic ones for risk-averse optimization and on the other hand, we prefer stochastic policies to avoid overfitting to the fixed batch of data. To this end, we use a similar parameterization to Fujimoto et al. (2019) and decompose the actor in two components: an imitation learning component $\pi^{\text{IL}}$ and a perturbation model $\xi_\theta$ such that the policy is expressed as:

$$\pi_\theta(s) = b + \lambda \xi_\theta(\cdot|s, b), \qquad \text{s.t., } b \sim \pi^{\text{IL}}(\cdot|s). \tag{8}$$

That is, $b$ is an action sampled from the imitation learning component, $\xi_\theta$ is a conditionally deterministic perturbation model that is optimized maximizing the actor loss (5) and $\lambda$ is a hyper-parameter that scales the perturbation magnitude. Thus, all the randomness in our policy arises from the behaviour policy and not from the subsequent optimization. Furthermore, if we would have access to the behaviour policy $\pi_\beta$ we could directly replace the imitation learning module by $\pi_\beta$.

To learn a generative model of the $\pi^{\text{IL}}$ from state-action pairs from the behaviour distribution $d^\beta$ we use a conditional variational autoencoder (VAE) (Kingma & Welling, 2014), which is also done in Fujimoto et al. (2019). The main advantage of the VAE compared to behavioral cloning (Bain & Sammut, 1995) is that it does not suffer from mode-collapse which hinders the actor optimization, and compared to inverse imitation learning (Ziebart et al., 2008; Abbeel & Ng, 2004) it does not assume that the policy is optimal. We chose the VAE over Generative Adversarial Networks (Ho & Ermon, 2016) due to easiness of training (Arjovsky & Bottou, 2017).

The conditional variational autoencoder is a probabilistic model that samples an action $b \sim \text{VAE}_\phi(s, a)$ according to the generative model

$$\mu, \Sigma = E_{\phi_1}(s, a); \qquad z \sim \mathcal{N}(\mu, \Sigma); \qquad b = D_{\phi_2}(s, z), \tag{9}$$

where $E_{\phi_1}$ is the neural network encoder, $D_{\phi_2}$ is the neural network decoder, and we sample the code $z$ using the re-parameterization trick (Kingma & Welling, 2014). To learn the $\phi = \{\phi_1, \phi_2\}$ parameters we place a prior $\mathcal{N}(0, I)$ on the code $z$ and minimize the variational lower-bound

$$\mathcal{L}_{\text{VAE}}(\phi) = \mathbb{E}_{s, a \sim \beta(\cdot)} \left[ \underbrace{(a - D_{\phi_2}(s, z))^2}_{\text{reconstruction loss}} + \frac{1}{2} \underbrace{\text{KL}(\mathcal{N}(\mu, \Sigma), \mathcal{N}(0, I))}_{\text{regularization}} \right]. \tag{10}$$

Upon a new state $s$, the generative model of the $\text{VAE}_\phi(s)$ is $z \sim \mathcal{N}(0, I)$, $b = D_{\phi_2}(s, z)$.

### 3.4 FINAL ALGORITHM

---

**Algorithm 1:** Offline Risk-Averse Actor Critic (O-RAAC).

---

**input** Data set, Critic $Z_w$ and critic-target $Z_{w'}$, $\text{VAE}_\phi = \{E_{\phi_1}, D_{\phi_2}\}$, Perturbation model $\xi_\theta$ and target $\xi_{\theta'}$, modulation parameter $\lambda$, Distortion operator $\mathcal{D}$ or distortion sampling distribution $\mathbb{P}_{\mathcal{D}}$, critic-loss parameters $N, N', \kappa$, mini-batch size $B$, learning rate $\eta$, soft update parameter $\mu$.
    **for** $t = 1, \dots$ **do**
        Sample $B$ transitions $(s, a, r, s')$ from data set.
        Sample $N$ quantiles $\tau$ and $N'$ target quantiles $\tau'$ from $\mathcal{U}(0, 1)$ and compute $\delta_{\tau, \tau'}$ in (2).
        Compute policy $\pi_\theta = b + \lambda \xi_\theta(s, b)$, s.t. $b \sim \text{VAE}_\phi(s, a)$ as in (9).
        Compute critic loss $\mathcal{L}_{\text{critic}}(w)$ in (4); actor loss $\mathcal{L}_{\text{actor}}(\theta)$ in (5); VAE loss $\mathcal{L}_{\text{VAE}}(\phi)$ in (10).
        Gradient step $w \leftarrow w - \eta \nabla \mathcal{L}_{\text{critic}}(w)$; $\theta \leftarrow \theta - \eta \nabla \mathcal{L}_{\text{actor}}(\theta)$; $\phi \leftarrow \phi - \eta \nabla \mathcal{L}_{\text{VAE}}(\phi)$.
        Perform soft-update on $w' \leftarrow \mu w + (1 - \mu) w'$; $\theta' \leftarrow \mu \theta + (1 - \mu) \theta'$.
    **end for**

---

We now combine the critic in Section 3.1, the actor in Section 3.2 and the VAE in Section 3.3 and show the pseudo-code of O-RAAC in Algorithm 1. We replace the expectations in the critic loss (4), actor loss (5) and VAE loss (10) with empirical averages of samples from the data set. As an ablation, we also propose the RAAC algorithm, in which the actor is parameterized with a neural network and there is no imitation learning component.

The critic's goal is to learn the reward distribution, the VAE goal is to learn a baseline action for the actor and the goal of the perturbation model is to be risk-averse. Although Santara et al. (2018) and Lacotte et al. (2019) propose risk-averse imitation learning algorithms, these interact with the environment in an on-policy way. Furthermore, the goal of the imitation learning component in O-RAAC is not to be risk-averse, but to provide a baseline to the risk-averse perturbation.

O-RAAC requires a distortion metric $\mathcal{D}$ as an input. For different $\mathcal{D}$, it generalizes existing distributional RL algorithms and extends them to the offline setting. For example, when $\mathcal{D}$ is the expectation operator, then the agent is risk neutral and O-RAAC is an offline version of D4PG (Barth-Maron et al., 2018). Likewise, when $\mathcal{D}$ is the Rockafellar-truncation operator (Rockafellar & Uryasev, 2002) we recover the algorithm by Singh et al. (2020) for optimizing the CVaR.

Table 1: Results of RAAC, WCPG, and D4PG in the car example. RAAC learns a policy that saturates the velocity before the risky region. WCPG and D4PG learn to accelerate as fast as possible, reaching the goal first with highest average returns but suffer from events with large penalty. We report mean (standard deviation) of each quantity.

| Algorithm | $CVaR_{0.1}$ | Mean | Risky Steps | Total Steps |
|-----------|-----------|------|-------------|-------------|
| **RAAC** | **48.0 (8.3)** | 48.0 (8.3) | **0 (0)** | 33 (1) |
| WCPG | 15.8 (3.3) | **79.8 (1.3)** | 13 (0) | **24 (0)** |
| D4PG | 15.6 (4.4) | **79.8 (2.0)** | 13 (0) | **24 (0)** |

## 4 EXPERIMENTAL EVALUATION

In this section, we test the performance of O-RAAC using $\mathcal{D} = CVaR_{\alpha=0.1}$ as risk distortion. We use Acerbi's formula (7) to compute the risk distortion. In particular, we ask:

1. How does RAAC perform as a *risk-averse* agent in the **off-policy** setting? (Section 4.1)

2. How does O-RAAC perform as a *risk-averse* agent in the **offline** setting? (Section 4.2)

3. How does O-RAAC perform as a *risk-neutral* agent in the **offline** setting? (Section 4.3)

For further details such as hyperparameter selection and extended results please refer to Appendix A.

### 4.1 OFF-POLICY SETTING: RISK-AVERSE PERFORMANCE

In this experiment, we intend to demonstrate the effectiveness of our algorithm as a risk-averse learner without introducing the extra layer of complexity of the offline setting.

#### 4.1.1 EXPERIMENTAL SETUP

As a toy example, we chose a 1-D car with state $s = (x, v)$, for position and velocity. The agent controls the car with an acceleration $a \in [-1, 1]$. The car dynamics with a time step $\Delta t = 0.1$ is

$$x_{t+1} = x_t + v_t \Delta t + 0.5 a_t (\Delta t)^2, \qquad v_{t+1} = v_t + a_t \Delta t.$$

The control objective is to move the car to $x_g = 2.5$ as fast as possible, starting from rest. To model the risk of crashing or of getting a speed fine, we introduce a penalization when the car exceeds a speed limit ($v > 1$). Hence, we use a random reward function given by

$$R_t(s, a) = -10 + 370 \mathbb{I}_{x_t = x_g} - 25 \mathbb{I}_{v_t > 1} \cdot \mathcal{B}_{0.2},$$

where $\mathbb{I}$ is an indicator function and $\mathcal{B}_{0.2}$ is a Bernoulli Random Variable with probability $p = 0.2$. The episode terminates after 400 steps or when the agent reaches the goal.

**Benchmarks** We compare RAAC with a risk-neutral algorithm (D4PG algorithm by Barth-Maron et al. (2018) with IQN and 1-step returns) and WCPG by Tang et al. (2020), a competing risk-averse algorithm. Both RAAC and WCPG optimize the 0.1-CVaR of the returns.

#### 4.1.2 RESULT DISCUSSION

In Table 1, we show the results of the experiment. As expected, D4PG ignores the low probability penalties and learns to accelerate the car with maximum power. Thus it has the largest expected return but the lowest CVaR. WCPG also fails to maximize the CVaR as it assumes that the return distribution is Gaussian. In turn, it under-estimates the variance of the return distribution of the maximum acceleration policy. Consequently, it over-estimates the CVaR of the returns and prefers the latter policy over the maximum-CVaR policy. In contrast, RAAC learns the full value distribution $Z$ and computes the CVaR reliably. Thus, it learns to saturate the velocity below the $v = 1$ threshold and finds the highest CVaR policy. See Appendix A.1 for more experiments in this setting.

**Qualitative Evaluation** In Figure 3 in Appendix A.1 we show the different trajectories as an illustration of the resulting risk-averse behavior.

## 4.2 Offline Setting: Risk-Averse performance

In this experiment, we intend to demonstrate the effectiveness of our algorithm as a risk-averse learner in the offline setting and in high-dimensional environments.

### 4.2.1 Experimental Setup

We test the algorithm on a variety of continuous control benchmark tasks on the data provided in the D4RL dataset (Fu et al., 2020). We use three MuJoCo tasks: HalfCheetah, Walker and Hopper (Todorov et al., 2012). In particular, we chose the medium (M) and expert (E) variants of these datasets. Since the tasks are deterministic, we incorporate stochasticity into the original rewards to have a meaningful assessment of risk and to showcase a practical example of when the risk-averse optimization makes sense. As a risk-averse performance metric, we chose the 0.1-CVaR of the episode returns. We use the following reward functions:

**Half-Cheetah:** $R_t(s,a) = \bar{r}_t(s,a) - 70\mathbb{I}_{v>\bar{v}} \cdot \mathcal{B}_{0.1}$, where $\bar{r}_t(s,a)$ is the original environment reward, $v$ the forward velocity, and $\bar{v}$ is a threshold velocity ($\bar{v} = 4$ for the (M) variant and $\bar{v} = 10$ for the (E) variant). As with the car example, this high-velocity penalization models a penalty to the rare but catastrophic event of the robot breaking – we want to be risk-averse to it. We evaluate the Half-Cheetah for 200 time steps.

**Walker2D/Hopper:** $R_t(s,a) = \bar{r}_t(s,a) - p\mathbb{I}_{|\theta|>\bar{\theta}} \cdot \mathcal{B}_{0.1}$, where $\bar{r}_t(s,a)$ is the original environment reward, $\theta$ is the pitch angle, $\bar{\theta}$ is a threshold angle ($\bar{\theta} = 0.5$ for the Walker2d-M/E and $\bar{\theta} = 0.1$ for the Hopper-M/E) and $p = 30$ for the Walker2d-M/E and $p = 50$ for the Hopper-M/E. When $|\theta| > 2\bar{\theta}$ the robot falls, the episode terminates, and we stop collecting such rewards. To avoid such situation, we shape the rewards with the stochastic event at $\theta > \bar{\theta}$. The maximum duration of the Walker2D and the Hopper is 500 time steps.

**Benchmarks** We optimize O-RAAC for the 0.1-CVaR. To demonstrate its effectiveness optimizing other risk distortions, we optimize O-RAAC using the 0.25-CVaR and the CPW distortion proposed by Tversky & Kahneman (1992); Gonzalez & Wu (1999). As a competing risk-averse algorithm, we augment WCPG with a VAE to yield the O-WCPG algorithm. As state-of-the-art risk-neutral agents, we use BEAR (Kumar et al., 2019) and O-D4PG, which is equivalent to BCQ (Fujimoto et al., 2019) with a distributional critic. As ablations, we compare RAAC (i.e. O-RAAC without the VAE) and the performance of the VAE as a pure imitation learning algorithm. Finally, we estimate the returns of the behavior policy by evaluating the stochastic reward functions on the data set and estimate its variance by bootstrapping batches of transitions.

### 4.2.2 Result Discussion

In Table 2, we show the results of the experiments. In all environments, O-RAAC has a higher 0.1-CVaR than the benchmarks. In environments that terminate, it also has longer duration than competitors. When optimizing for other risk-averse criteria, O-RAAC still has good 0.1-CVaR performance. The behavior policy has poor risk-averse performance and the VAE ablation imitates such policy, hence performs poorly too. On the other side of the spectrum, RAAC performs poorly in all categories. This indicates that the offline version enhancement of RAAC is crucial for offline problems. O-WCPG performs poorly in terms of the CVaR. This might be due to the return distribution not being Gaussian. Finally, BEAR and O-D4PG usually have good risk-neutral performance, particularly in the medium version of these datasets, but poor risk-averse performance. This is expected as these algorithms are designed to optimize the risk-neutral performance.

**Qualitative Evaluation** We visualize the risk-averse performance by looking at the support of the risk-event in Figure 2. Most of the support of the data set induced by the behavior policy lies in the risky region. O-RAAC learns to shift the distribution towards the risk-free region (green shaded area). On the other hand, O-D4PG (risk-neutral) struggles to shift the distribution as it ignores the rare penalties in the risky region. Only in the Half Cheetah-E experiment O-D4PG manages to shift the distribution towards the safe region. This is because most of the behavior policy is on the risky region and the *average* performance of the behavior policy under the new stochastic rewards is low.

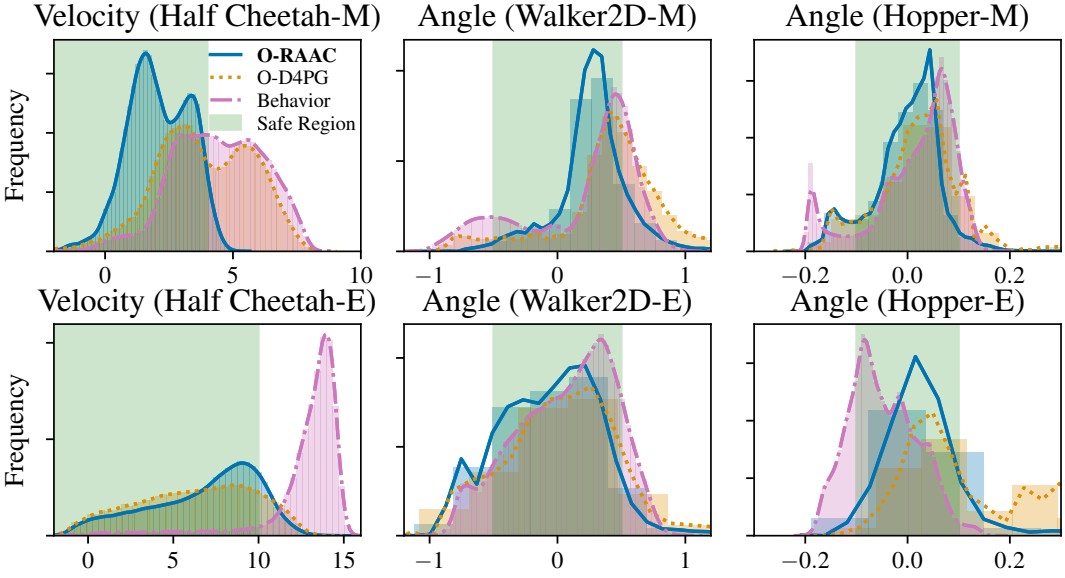

Figure 2: Empirical distribution of the risk-event of O-RAAC, O-D4PG, and the behavior policies. O-RAAC shifts the support towards the risk-free region (green area). On the other hand, O-D4PG ignores the risk-related rewards and imitates the behavior distribution.

## 4.3 OFFLINE SETTING: RISK-NEUTRAL PERFORMANCE

It is well-known that coherent risk-related criteria have a dual distributional robust criterion formulation (Shapiro et al., 2014; Iyengar, 2005; Osogami, 2012). In particular, the following holds:

$$\max_{\pi} \mathcal{D}\left[Z^\pi(x,a)\right] = \max_{\pi} \min_{d \in \overline{\mathcal{D}}_\pi} \mathbb{E}_d\left[Z^\pi(x,a)\right], \tag{11}$$

where $\overline{\mathcal{D}}_\pi$ is a dual set of distribution that is induced by the distortion measure $\mathcal{D}$ and the distribution $d_\pi$. When the distortion set is the expectation, the dual set collapses to a singleton $\overline{\mathcal{D}}_\pi = \{d^\pi\}$. For the CVaR, Chow & Ghavamzadeh (2014) expresses the dual set for MDPs in Proposition 1. Given this dual result, it is straightforward to show that $\mathbb{E}_{d^\pi}[Z^\pi(x,a)] \geq \mathcal{D}[Z^\pi(x,a)]$ by definition of the minimum. In this sense, optimizing $\mathcal{D}[Z^\pi(x,a)]$ is equivalent to optimizing a *pessimistic* estimate of the risk-neutral performance, in a similarly way to Buckman et al. (2020) and Kumar et al. (2020).

### 4.3.1 EXPERIMENTAL SETUP

Despite the goal being to *maximize a risk-neutral objective*, we evaluate whether it is beneficial to *optimize a risk-averse criterion* in the offline setting. We test this hypothesis using the same setup as in Section 4.2. Namely, we train O-RAAC using a risk-averse metric, but we evaluate on the average value, a risk-neutral metric.

**Benchmarks** We use the same benchmarks as in Section 4.2.

### 4.3.2 RESULT DISCUSSION

We show the risk-neutral performance in the "Mean" columns in Table 2. In all data sets, O-RAAC performs better or similar to the benchmarks. A reason for this is that the episodes of risk-averse agents last longer and thus collect rewards for more time steps. However, this is not the *only* reason. For example, BEAR in the Walker-expert environment has longer episodes and still lower mean returns than O-RAAC. In the Hopper-expert, RAAC and VAE have similar duration to O-RAAC. Yet, O-RAAC has a higher average return than each of the ablations. This indicates that optimizing a risk-averse performance is beneficial when comparing the risk-neutral performance, specially in data sets where the data is not very diverse (e.g., in the "expert" data sets).

**Qualitative Evaluation**    In Figure 2, we see that in most cases there *is* a distribution shift between the behaviour distribution and both O-RAAC and O-D4PG. As the shift increases, we see the benefits of learning using distributionally robust objectives. As a particular example we take the Hopper-E distribution. The behavior policy is safe as it does not terminate before the 500 time steps but much of the data is on the risky region. O-RAAC learns to center the distribution in the safe region and yet hops forward efficiently. On the other hand, O-D4PG also learns to shift away of the risky region. However, it is not risk averse and it overshoots towards the other end of the risky region, where there is not sufficient data to have good critic estimates.

Table 2: Performance metrics on offline MuJoCo data sets with medium (first column block) and expert (second column block) datasets. We compare the 0.1-CVaR and mean of the episode returns, and the average episode duration. We report the mean (standard deviation) of metric. In all environments, O-RAAC has a higher CVaR than benchmarks. In environments that terminate, O-RAAC has a longer duration too. Finally, O-RAAC has comparable risk-neutral performance to benchmarks.

| | Algorithm | Medium | | | Expert | | |
| | | $CVaR_{0.1}$ | Mean | Duration | $CVaR_{0.1}$ | Mean | Duration |
|---|---|---|---|---|---|---|---|
| Half-Cheetah | **O-RAAC$_{0.1}$** | **214 (36)** | **331 (30)** | 200 (0) | **595 (191)** | **1180 (78)** | 200 (0) |
| | **O-RAAC$_{0.25}$** | **252 (14)** | **317 (5)** | 200 (0) | **695 (34)** | **1185 (7)** | 200 (0) |
| | **O-RAAC$_{CPW}$** | **253 (9)** | **318 (3)** | 200 (0) | 358 (67) | 974 (21) | 200 (0) |
| | O-WCPG | 76 (14) | **316 (23)** | 200 (0) | 248 (232) | **905 (107)** | 200 (0) |
| | O-D4PG | 66 (34) | **341 (20)** | 200 (0) | 556 (263) | 1010 (153) | 200 (0) |
| | BEAR | 15 (30) | **312 (20)** | 200 (0) | 44 (20) | 557 (15) | 200 (0) |
| | RAAC | -55 (1) | -52 (0) | 200 (0) | 3 (13) | 30 (3) | 200 (0) |
| | VAE | 10 (23) | **354 (9)** | 200 (0) | 260 (84) | 754 (18) | 200 (0) |
| | Behavior | 9 (6) | **344 (2)** | 200 (0) | 100 (8) | 727 (4) | 200 (0) |
| Walker-2D | **O-RAAC$_{0.1}$** | **751 (154)** | **1282 (20)** | 397 (18) | **1172 (71)** | **2006 (56)** | **432 (11)** |
| | **O-RAAC$_{0.25}$** | 497 (71) | **1257 (27)** | **479 (6)** | 670 (133) | 1758 (48) | **436 (7)** |
| | **O-RAAC$_{CPW}$** | 500 (71) | **1304 (16)** | **477 (3)** | 819 (89) | 1874 (34) | **454 (8)** |
| | O-WCPG | -15 (41) | 283 (37) | 185 (12) | 362 (33) | 1372 (160) | 301 (31) |
| | O-D4PG | 31 (29) | 308 (20) | 249 (9) | 773 (55) | 1870 (63) | 405 (12) |
| | BEAR | 517 (66) | **1318 (31)** | **468 (8)** | 1017 (49) | 1783 (32) | **463 (4)** |
| | RAAC | 55 (2) | 92 (9) | 200 (7) | 54 (2) | 83 (6) | 196 (6) |
| | VAE | -84 (21) | 425 (37) | 246 (9) | 345 (302) | 1217 (180) | **350 (130)** |
| | Behavior | -56 (9) | 727 (16) | 500 (0) | 1028 (34) | 1894 (7) | 500 (0) |
| Hopper | **O-RAAC$_{0.1}$** | **1416 (28)** | 1482 (4) | **499 (1)** | **980 (28)** | **1385 (33)** | **494 (6)** |
| | **O-RAAC$_{0.25}$** | 1108 (14) | 1337 (21) | 419 (6) | **730 (129)** | 1304 (21) | 434 (6) |
| | **O-RAAC$_{CPW}$** | 969 (9) | 1188 (6) | 373 (2) | 488 (1) | 496 (0) | 160 (0) |
| | O-WCPG | -87 (25) | 69 (8) | 100 (0) | 720 (34) | 898 (12) | 301 (1) |
| | O-D4PG | 1008 (28) | 1098 (11) | 359 (3) | 606 (31) | 783 (18) | 268 (3) |
| | BEAR | 1252 (47) | **1575 (8)** | 481 (2) | 852 (30) | 1180 (12) | 431 (4) |
| | RAAC | 71 (23) | 113 (5) | 146 (4) | 474 (0) | 475 (0) | **500 (0)** |
| | VAE | 727 (39) | 1081 (17) | 462 (4) | 774 (36) | 1116 (13) | **498 (1)** |
| | Behavior | 674 (5) | 1068 (4) | 500 (0) | 827 (12) | 1211 (3) | 500 (0) |

## 5  CONCLUSION

In high-stakes applications, decision-making is usually risk-averse and no interactions with the environment are allowed. For this practical setting, we introduce O-RAAC, the first fully offline risk-averse algorithm. O-RAAC is compatible with many common risk-averse criteria such as coherent-risk measures or cumulative prospect theory. Due to the distributional-robust properties of risk-sensitive criteria, it also optimizes risk-neutral criteria under natural distribution shift that occur in the offline setting. Empirically, O-RAAC outperforms other algorithms in terms of risk-averse performance and is competitive with risk-neutral algorithms in terms of risk-neutral performance. Particularly, in cases where there is not much data diversity, such as in expert data sets, optimizing risk-averse metrics is beneficial due to inherent robustness properties.

ACKNOWLEDGMENTS AND DISCLOSURE OF FUNDING

This project has received funding from the European Research Council (ERC) under the European Unions Horizon 2020 research and innovation program grant agreement No 815943.

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

## A  EXTENDED EXPERIMENTAL RESULTS

### A.1  CAR

We ran the Car environment for RAAC, D4PG and WCPG using 5 independent random seeds. We evaluate final policies for 1000 interactions and report the averaged results with corresponding standard deviation in Table 1. In Figure 3, we show the trajectories when following aforementioned policies.

**Reward function design**  We use the reward function given by

$$R_t(s, a) = -10 + 370\mathbb{I}_{x_t = x_g} - 25\mathbb{I}_{v_t > 1} \cdot \mathcal{B}_{0.2},$$

where $\mathbb{I}$ is an indicator function and $\mathcal{B}_{0.2}$ is a Bernoulli Random Variable with probability $p = 0.2$. That is, $r_f = 370$ is a sparse reward that the agent gets at the goal and $r_d = -10$ is a negative reward that penalizes delays on reaching the goal. Finally, the agent receives a negative reward of $r_v = -25$ with probability 0.2 when it exceeds the $v > 1$ threshold. As the returns is a sum of bernoulli R.V. we know that it will be a Binomial distribution. For this particular case, we expect that if the number of steps is large enough, the Gaussianity assumption that WCPG does is good as Binomial distributions are asymptotically Gaussian (Vershynin, 2018). However, the episode terminates after at most thirteen risky steps and the approximation is not good.

We show in Figure 3 the trajectories for RAAC, D4PG and WCPG.

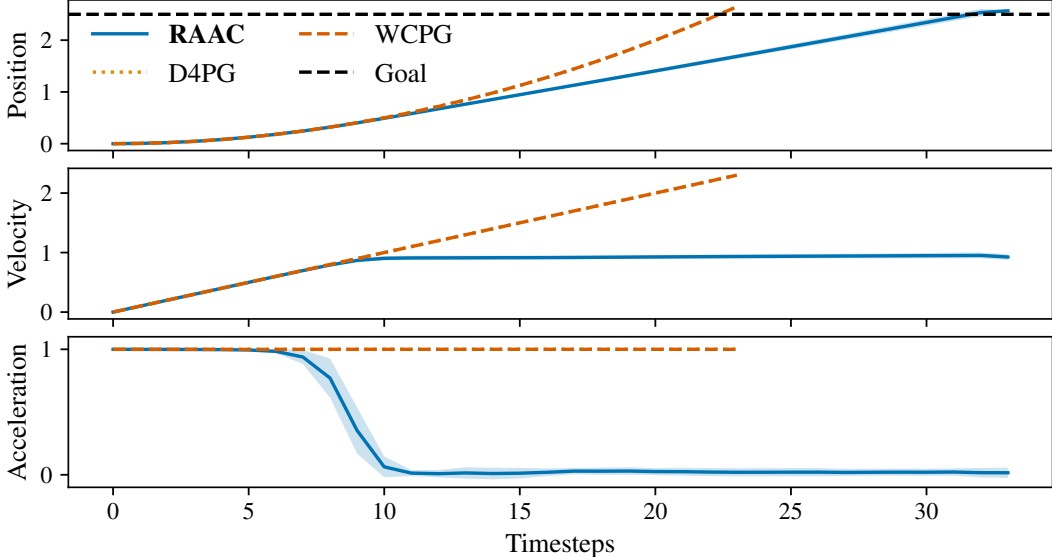

Figure 3: Evolution of car states and input control when following learned policies for RAAC, WCPG and D4PG. We use policies from 5 independent seeds for each algorithm. RAAC learns to saturate the velocity below the speed limit.

## A.2 MuJoCo Environments

We ran 5 independent random seeds and evaluate for 20 episodes the policy every 100 gradient steps for *HalfCheetah* and 500 gradient steps for *Hopper* and *Walker2d*. We plot the learning curves of the medium variants in Figure 4 and expert variant in Figure 5. To report the tests in Table 2, we early-stop the policy that outputs the best CVaR and evaluate on 100 episodes with 5 different random seeds.

**Behavior policies** For sake of reference, we evaluate the stochastic reward function on the state-action pairs in the behavior data set. Unfortunately, the data sets do not distinguish between episodes. Hence, to estimate the returns, we use the state-action distribution in the data set and split it into chunks of 200 time steps for the Half-Cheetah and 500 time steps for the Walker2D and the Hopper. We then compute the return of every chunk by sampling a realization from its stochastic reward function. Finally, we bootstrap the resulting chunks into 10 datasets by sampling uniformly at random with replacement and estimate the mean and $\text{CVaR}_{0.1}$ of the returns in each batch. We report the average of the bootstrap splits together with the standard deviation in Table 2.

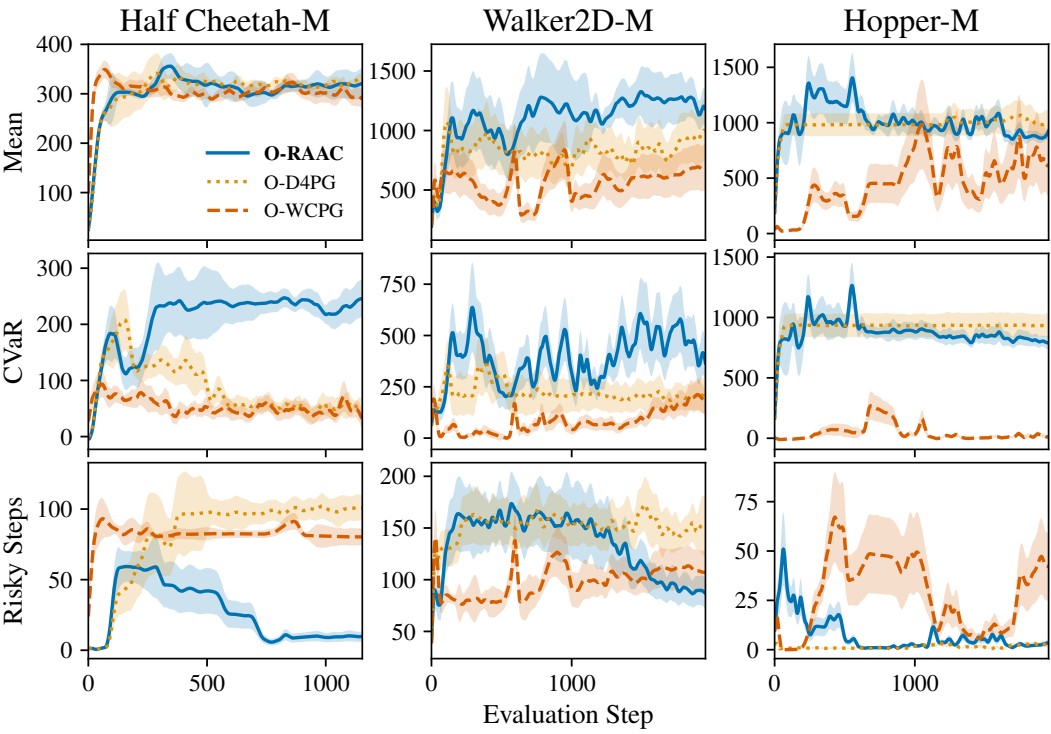

Figure 4: Experimental results across several Mujoco tasks for the Medium variant of each dataset.

## A.3 Additional Experimental Details

### A.3.1 Architectures

We use neural networks as function approximators for all the elements in the architecture.

**Critic architecture:** For the critic architecture, we build on the IQN network Dabney et al. (2018) but we extend it to the continuous action setting by adding an additional action input to the critic network, resulting in the function:

$$Z(s, a; \tau) = f(m_{sa\tau}([m_{sa}([\psi_s(s), \psi_a(a)]), \psi_\tau(\tau)]),$$ (12)

where $\psi_s : \mathcal{X} \to \mathbb{R}^d$, $\psi_a : \mathcal{A} \to \mathbb{R}^d$, $m_{sa} : \mathbb{R}^{d \times d} \to \mathbb{R}^n$, $m_{sa\tau} : \mathbb{R}^n \to \mathbb{R}^n$, $\psi_\tau : \mathbb{R} \to \mathbb{R}^n$ and $f : \mathbb{R}^n \to \mathbb{R}$

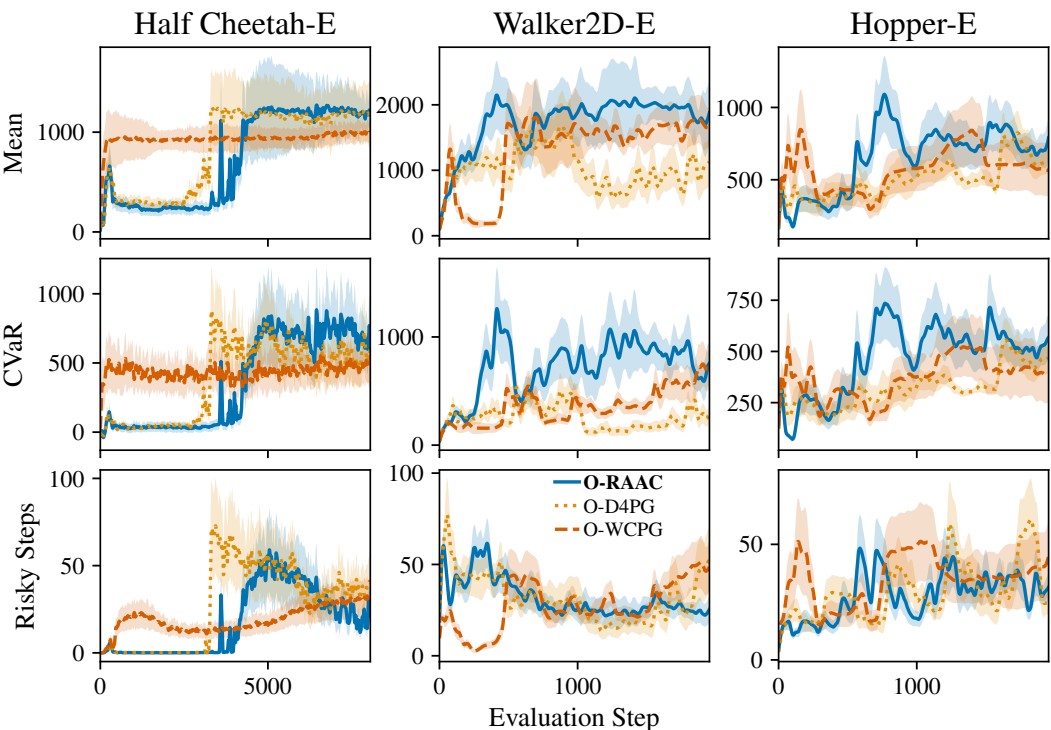

Figure 5: Experimental results across several Mujoco tasks for the Expert variant of each dataset.

For the embedding $\psi_\tau$ we use a linear function of $n$ cosine basis functions of the form $\cos(\pi i \tau)$ $i = 1.., n$, with $n = 16$ as proposed in Dabney et al. (2018). For $\psi_s, \psi_a$ we use a multi-layer perceptron (MLP) with a single hidden layer with $d = 64$ units for the Car experiment and with $d = 256$ units for all MuJoCo experiments. For the merging function $m_{sa}$, which takes as an input the concatenation of $\psi_s(s)$ and $\psi_a(a)$, we use a single hidden layer with $n = 16$ units. For the merging function $m_{sa\tau}$, we force interaction between its two inputs via a multiplicative function $m_{sa\tau}(u_1, u_2) = u_1 \odot u_2$, where $u_1 = m_{sa}(\psi_s(s), \psi_a(a))$ and $u_2 = \psi_\tau(\tau)$ and $\odot$ denotes the element-wise product of two vectors. For $f$ we use a MLP with a single hidden layer with 32 units We used ReLU non-linearities for all the layers.

**Actor architecture:**  The architecture of the actor model is

$$\pi(a|s) = b + \lambda \xi_\theta(s, b) \tag{13}$$

where $\xi : \mathcal{A} \to \mathbb{R}^{\|\mathcal{A}\|}$ and $b$ is the output of the imitation learning component. For the RAAC algorithm we remove $b$ and set $\lambda = 1$.

For the Car experiments, we used a MLP with 2 hidden layers of size 64. For the MuJoCo experiments, based on Fujimoto et al. (2019), we used a MLP embedding with 3 hidden layers of sizes 400, 300 and 300. We used ReLU non-linearities for all the hidden layers and we saturate the output with a Tanh non-linearity.

**VAE architecture:**  The architecture of the conditional VAE$_\phi$ is also based on Fujimoto et al. (2019). It is defined by two networks, an encoder $E_{\phi_1}(s, a)$ and decoder $D_{\phi_2}(s, z)$. Each network has two hidden layers of size 750 and it uses ReLU non-linearities.

### A.3.2 HYPERPARAMETERS

All the network parameters are updated using Adam (Kingma & Ba, 2015) with learning rates $\eta = 0.001$ for the critic and the VAE, and $\eta = 0.0001$ for the actor model, as in Fujimoto et al. (2019). The target networks for the critic and the perturbation models are updated softly with $\mu = 0.005$.

For the critic loss (4) we use $N = N' = 32$ quantile samples, whereas to approximate the CVaR to compute the actor loss (5) (7) we use 8 samples from the uniform distribution between $[0, 0.1]$.

In Figure 6, we show an ablation on the effect of the hyper-parameter lambda. As we can see, a correct selection of lambda is of crucial performance as it trades-off pure imitation learning with pure reinforcement learning. As $\lambda \to 0$, the policy imitates the behavior policy has poor risk-averse performance. As $\lambda \to 1$, the policy suffers from the bootstrapping error and the performance is also low. We find values of $\lambda \in [0.05, 0.5]$ to be the best, although the specific $\lambda$ is environment dependent. This observation coincides with those in Fujimoto et al. (2019, Appendix D.1).

For all MuJoCo experiments, the $\lambda$ parameter which modulates the action perturbation level was experimentally set to 0.25, except for the HalfCheetah-medium experiment for which it was set to 0.5. As we can see from Figure 6, this is not the best value of $\lambda$, but rather a value that performs well across most environments.

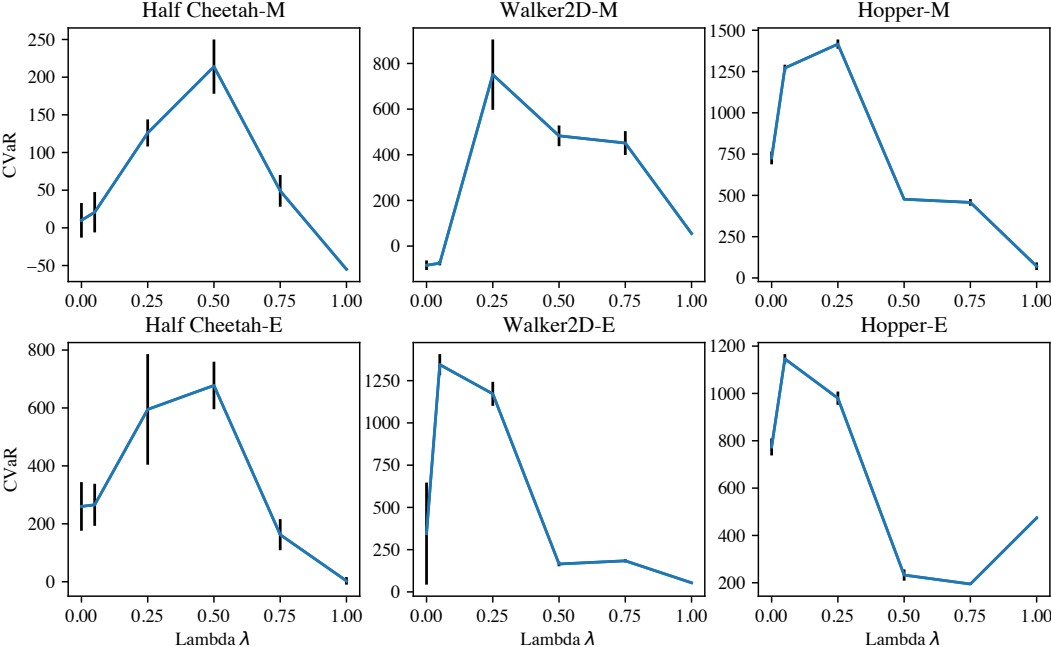

Figure 6: Effect of the hyperparameter $\lambda$ on the CVaR of the returns for each of the MuJoCo environments. As $\lambda \to 0$, the policy imitates the behavior policy has poor risk-averse performance. As $\lambda \to 1$, the policy suffers from the bootstrapping error and the performance is also low. The best $\lambda$ is environment dependent.

