# OpenReview forum: "Risk-Averse Offline Reinforcement Learning"
_ICLR.cc/2021/Conference — ICLR 2021 Poster_

### Official Review · AnonReviewer4 · 2020-10-28
**An interesting approach that put together state-of-the-art techniques in order to obtain a strong risk-averse offline learner**

**Rating:** 6
**Confidence:** 4

**Review:**

## Brief Summary
The authors developed an risk-averse RL algorithm with operates in offline contexts. The main idea consists in putting together the benefits from 3 previous works in the distributional, risk-averse and offline RL fields:
- the DDPG distributional extension (Barth-Maron et al., 2018), called D4PG
- the Implicit Quantile Network (Dabney et al., 2018), called IQN
- actor using an imitation learning component with perturbation to control the bootstrappinge error (Fujimoto et al. 2019)
The online version of the resulting algorithm, called RAAC, is then tested on a toy problem.
Later, the full algorithm (O-RAAC) is tested on 3 MuJoCo tasks, using offline data from the the D4RL dataset (Fu et al., 2020).
The performance is analysed on both a risk-averse and risk-neutral point of view.

## Strong Points
- The state-of-the-art is correctly reported.
- The paper is clear and easy to follow.
- The experimental analysis is correct, complete and interesting:
    + baselines seem to be competitive: they are also augmented with the VAE layer
    + they  the analysis is sound and provides interesting insights.
- The proposed approach seems to outperform the baselines on the optimized measure: moreover, as noticed in also in previous work, introducing risk-aversion seems to be useful also to increase the expected return.

## Weak Points

- The algorithm is obtained by composing existing approach: it is not clear to me whether there is a novel methodological contribution or not.
- The authors claim that their approach is general for any risk-measure, however, the experiments are conducted only for the CVaR case and with a specific percentile.
- The results in the appendix does not seem to match the ones in the main paper.

## Recommendation
My reccomendation is to accept the paper since it provides an interesting approach which puts together, in an original way, techniques from different areas of RL, even if it is not clear whether there is a novel methodological contribution. The shown results seems to be relevant, however, it is still not clear whether they are

## Questions for the authors
- Performances seem to be worse in the supplementary material, can you explay why?
- Could you provide experiments with different quantiles for CVaR?
- Could you provide experiments with different risk-measures?
- Is it possible to evaluate in a qualitative way the behavior obtained by the agents in the learned task when using the risk-aversion or not?

## Additional Feedback
- It would be useful to state the dependecies of your approach in a clearer way.

---

> ### Author Response · Authors · 2020-11-24
> **Thank you for your review. See answers below.**
>
> We thank the reviewer for their valuable feedback and comments. We answer the points below:
>
> - Performance differences: thanks for catching this, it was a bug in our plotting code. We fixed the training curves in the rebuttal version.
>
> - Different quantiles for CVaR: Yes, we included experiments with $\alpha=0.25$ in Table 2.
>
> - Different risk-measures: Yes, we included experiments with a different risk measure (cumulative prospect weights) in Table 2.
>
> - Qualitative evaluation: Figure 3 in the appendix is a qualitative demonstration of the risk-averse behavior in a simple 1-d task. Figure 2 in the main paper is also a way of demonstrating this. We added a paragraph in each experiment with a "Qualitative Evaluation" title, followed by an appropriate discussion.

---

> > ### Comment · AnonReviewer4 · 2020-11-24
> > **Comment**
> >
> > Thank you for your feedback, some more comments:
> > - In Figure 3 in the appendix D4PG plot is missing
> > - thank you for adding the results optimizing the other quantile and CPW

---

> > > ### Author Response · Authors · 2020-11-24
> > > **Thank you for your comments. See answers below.**
> > >
> > > We thank the reviewer for their valuable feedback and comments.
> > >
> > > Regarding the Figure 3 in appendix, the D4PG plot is actually overlapped by the WCPG trajectory, since both trajectories are exactly the same. We do agree that it can lead to confusion and we will clarify that in the camera ready version of the paper.

---

### Official Review · AnonReviewer2 · 2020-10-28
**Recommendation to Accept**

**Rating:** 8
**Confidence:** 3

**Review:**

##########################################################################

Summary:

This paper proposes O-RAAC, an offline RL algorithm that minimizes the Conditional Value-at-Risk (CVaR) of the learned policy's return given  a dataset by a behavior policy. It learns a distributional critic, a VAE for imitation learning, and an actor that perturbs the VAE output to minimize the risk given by the distortion operator D (Here it uses CVaR).


Several experiments were performed to show the effectiveness of O-RAAC, both with a simple 1-d driving environment and with a modified version of D4RL dataset.



##########################################################################
Pros:

The paper is well written, with comparisons with competing methods throughout. This makes the connection with those methods clear and easy to understand. The discussion around the design choices and the tradeoffs are especially insightful. (e.g. Huber loss over l1 or l2; VAE vs vanilla BC; etc.)

The contribution to risk averse offline reinforcement learning is novel, with the use of CVaR and the idea of a perturbed version of a imitation learning actor. With the assumption that all the risks is captured within the reward.


##########################################################################

Suggestions:

The risk-neutral experiment in 4.3 was a bit confusing. I was not sure whether you measured the O-RAAC’s risk-neutral performance by setting alpha = 0, or you measured its performance with alpha=0.1, but using a risk-neutral metric. I think you did the latter after a few more reading. More clarification either in Table 2 or in 4.3 would be great.

in 3.3, you said "...is a perturbation model that is optimized maximizing the actor loss (5)," This part was confusing as well, since one would normally minimize a loss. In this paper, actor loss = risk aversion, so maximizing risk aversion makes sense, but I wouldn't call it a loss.

---

> ### Author Response · Authors · 2020-11-24
> **Thank you for your review. See answers below.**
>
> We thank the reviewer for their valuable feedback and comments. We answer the points below:
>
> - Risk-neutral experiment: We trained O-RAAC with alpha=0.1, but evaluate using the average return. We clarify this in the rebuttal version (Section 4.3.1). We also added other risk-measures as suggested by other reviewers (Table 2).
>
> - Actor loss nomenclature: We thank the reviewer for this helpful comment. We oversaw this when submitting the paper. The actor loss is the negative of such quantity. We changed this in the rebuttal version (equation 5 and last sentence in first paragraph, as well as in the algorithm)

---

### Official Review · AnonReviewer1 · 2020-10-29
**Encouraging empirical results. Unclear theoretical properties, hard to implement in new tasks.**

**Rating:** 5
**Confidence:** 4

**Review:**

The authors propose an RL algorithm for learning risk-averse policies from offline data. Empirically, it is shown that it can outperform some existing risk-neutral approaches on a number of challenging robotic control tasks under risk-sensitive performance measures. Although the empirical results are encouraging, the theoretical properties of the proposed algorithm are unclear and therefore it is not clear how easy it can be implemented in other tasks.

Overall, the paper is easy to read and the presentation is clear.

The authors address a very important issue that is faced by RL practitioners. Learning risk-averse policies in a fully offline setting is inherently ill-posed and the burden is in incorporating sufficiently strong prior and bias in regularizing the learning system. Unfortunately the current paper falls short in this regard, where the only idea here is to use imitation learning in the form of autoencoders. So the main contribution in terms of new idea is rather incremental.

Furthermore, the theoretical properties of the proposed algorithm are largely unclear. For example, with respect to a fixed distribution for the start state $S_1$, the optimal CVaR policy for Eq.(1) can be non-stationary. The authors restrict the policy search to the space of stationary deterministic policies. Now suppose that the neural network is over-parameterized, and assuming arbitrarily large training set, which policy does the algorithm converge to? Does the algorithm converge?

Considering the 3 gradient updates in Algorithm 1, one wonders how to choose all those parameters to make it work, and whether a single step-size parameter is enough. In particular, $\lambda$ seems hard to choose. How sensitive are the empirical results with respect to the choice of these parameters? In the truly offline case, one presumably can only perform fine-tuning using a separate validation set, how should this be done here?

---

> ### Author Response · Authors · 2020-11-24
> **Thank you for your review. See answers below.**
>
> We thank the reviewer for their valuable feedback and comments. We answer the points below:
>
> - Incremental contribution: we disagree with this claim:  we study a novel and highly relevant setting, and provide a novel method, which builds on and adapts state-of-the-art algorithms for this setting.
>
> - The optimal CVaR policy for Eq.(1) can be non-stationary: Although there are non-stationary policies that are optimal, also there exists a stationary policy that is optimal for many risk measures (see "Andrzej Ruszczýn .Risk-averse dynamic programming for Markov decision processes. Mathematical programming, 2010", Theorem 4). The CVaR is one of such risk measures. We agree that in the finite horizon setting, the optimal policy is non-stationary. We clarify this in the rebuttal version.
>
> - Convergence: As in most deep RL algorithms, the theory lacks behind. We did not claim any theoretical guarantees but believe that the empirical results are promising.
>
> - Hyperparameter selection: Indeed this is an interesting open problem for offline RL. We selected the default hyperparameters for BCQ. Whether these were originally chosen based on the true environment is unknown to us.
> We also provide an ablation experiment on the effect of $\lambda$ in Appendix A.3.

---

### Official Review · AnonReviewer3 · 2020-11-03
**Study the problem of safety in offline RL, and propose a novel algorithm for learning risk-averse policy**

**Rating:** 6
**Confidence:** 3

**Review:**

Summary
The paper studies the problem of safe reinforcement learning, where we want to learn risk-averse policies in the offline setting. It proposes “Offline Risk Averse Actor Critic” (ORAAC) which performs competitively as risk-neural agent, and outperforms D4PG based baseline as a risk-averse agent on D4RL benchmark. The algorithm involves modifying the losses to learn risk-averse actor, distributional critic and a VAE-based imitative policy.


Reasons for the score:
I vote in favour of accepting the paper. The paper studies safety aspects of learning algorithms which are competitive as a risk-averse agent in offlineRL. I would strongly encourage the authors to include numbers for baselines like CQL/BEAR. Though not risk-averse, these algorithms are conservative by design and are strong baselines in offlineRL.


Strengths:
+ The problem is well motivated and the ideas are presented clearly. The authors perform extensive experiments, ablations and provide empirical evidence where ORAAC outperform baselines like OD4PG on D4RL benchmark tasks.
+ Though risk-averse by design, the learned policy is competitive as a risk-neutral agent.

Weaknesses:
- Comparison to SOTA baselines in the risk-neutral scenario are missing. Though CQL/BEAR are not risk-averse my design, these algorithms are competitive in the risk-neutral setting. It would be critical to know how ORAAC performs in comparison to such baselines.
- Discussion on the choice of hyperparameters and experiment design are under-explored. Elaborating on this in section 4 would significantly improve the readability of the paper.

[1] CQL : https://arxiv.org/abs/2006.04779
[2] BEAR : https://arxiv.org/abs/1906.00949

---

> ### Author Response · Authors · 2020-11-24
> **Thank you for your review. See answers below.**
>
> We thank the reviewer for their valuable feedback and comments. We answer the points below:
>
> - Comparison to SOTA baselines: CQL was accepted to NeurIPS after the ICLR deadline. We include a comparison to BEAR in Table 2.
> Nonetheless, BCQ (O-D4PG with a distributional critic) is also a risk-neutral baseline as it is a comparison to a risk-neutral agent.
>
> - Experimental Design: We improved Section 4 and the appendix based on the suggestions of the reviewer.  We also added an ablation on the effect of $\lambda$ in Appendix A.3.

---

### Official Review · AnonReviewer5 · 2020-11-06
**Official Blind Review**

**Rating:** 7
**Confidence:** 3

**Review:**

The submission investigates the risk averse objective in offline RL. Usually, the parametric uncertainty is the main source of worries in offline RL, and dealing with the stochastic uncertainty on top of it, in order to account for risk aversion, is very challenging. The authors expose clearly their method and algorithm, even though we sometimes would have liked a bit more argumentation on why this and why not that. Since no theoretical analysis is provided, the only validation is empirical. It is rather complete regarding both the settings and the domains. The results are quite impressive, in particular in the offline setting. For all these reasons, I recommend to accept the submission.

Please answer/address these comments in the rebuttal/final version:
* 1- It is unclear whether d^\beta is defined as the empirical distribution in the batch, or the true distribution. I believe that it would have been helpful in general to formalize more the distributions definitions.
* 2- U(.) is not introduced.
* 3- First sentence of 3.2: It is frequent in offline RL to use stochastic policies in order to leverage the risk taken in the face of parametric uncertainty. It is therefore rather odd to state here that only deterministic policies are considered. Even more when we notice in 3.3 that the actual policy is stochastic, since b is sampled from the stochastic behavioral policy.
* 4- Please indicate the performance of the behavioural policy (average and cvar) in the experiments.

---

> ### Author Response · Authors · 2020-11-24
> **Thank you for your review. See answers below.**
>
> We thank the reviewer for their valuable feedback and comments. We answer the points below:
>
> 1- $d^\beta$ is the distribution induced by the behavior policy (and $\beta$ quantities). We access it by sampling from the batch. We clarified this in the rebuttal version (first paragraph from section 2).
> 2- $\mathcal{U}(\cdot)) is the uniform distribution. We clarified this in the rebuttal version (above equation 3).
> 3- deterministic vs stochastic policies: Thanks for this comment. We prefer to avoid stochastic policies because they are not optimal, they are an extra source of stochasticity, and in the offline setting, they bring no exploration benefits. We nevertheless agree that they help to avoid overfitting and that is what most works use. We clarified this in the rebuttal version (first two paragraphs from section 3.3).
> 4- We indicated the performance of the behavioral policy (Mean and CVaR) in the appendix (Table 3). We now include it in the main paper in the rebuttal version, together with uncertainty estimation via bootstrap (Table 2).
> We also add the performance of the VAE as an imitation learning benchmark.

---

### Decision · Program_Chairs · 2021-01-07
**Final Decision**

**Decision:**

Accept (Poster)

**Comment:**

This paper proposes O-RAAC, an offline RL algorithm that minimizes the Conditional Value-at-Risk (CVaR) of the learned policy's return given a dataset by a behavior policy. The reviews are generally positive with most agreeing that the paper presents interesting empirical results.

The experiments are limited to simpler domains, and could be extended to include harder tasks from other continuous control domains. Some examples could be domains such as in Robosuite (http://robosuite.ai/) or Robogym (https://github.com/openai/robogym). These environments have higher dimensional systems with more clearer safety settings.
Agreeably, asking for comparisons with unpublished results may be unfair, however, it would be recommended to authors to include additional comparisons with latest methods in Offline/Batch-RL, including the ones which don't guarantee risk, such as CQL, BRAC, CSC.

Further, The theoretical properties of the proposed algorithm are largely unclear. It would help to analyze the effect of both convergence rates, and fixed points, further what is the effect of addition of risk, does the algorithm converge to a suboptimal solution or get there slower. Finally empirical reporting of cumulative number of failures (discrete count) during training as well as during evaluation would be very useful to practitioners.

Other relevant and concurrent papers to potentially take note of:
Distributional Reinforcement Learning for Risk-Sensitive Policies (https://openreview.net/forum?id=19drPzGV691)
Conservative Safety Critics for Exploration (https://openreview.net/forum?id=iaO86DUuKi)

I would recommend acceptance of the paper. I would strong encourage release of sufficiently documented and easy to use implementation.  Given the fact that the main argument is empirical utility of the method, it would be limit the impact of this work if readers cannot readily build on O-RAAC.